

# Risk factors for new-onset atrial fibrillation in patients with chronic obstructive pulmonary disease: a systematic review and meta-analysis

Qiangru Huang, Huaiyu Xiong, Tiankui Shuai, Meng Zhang, Chuchu Zhang, Yalei Wang, Lei Zhu, Jiaju Lu and Jian Liu

Department of Intensive Care Unit, The First Hospital of Lanzhou University, Lanzhou, China
The First Clinical Medical College of the First Hospital of Lanzhou University, LanZhou, China

Corresponding author
Jian Liu, medecinliu@sina.com

## ABSTRACT

**Background**. New-onset atrial fibrillation (AF) in patients with chronic obstructive pulmonary disease (COPD) is associated with an accelerated decline in lung function, and a significant increase in mortality rate. A deeper understanding of the risk factors for new-onset AF during COPD will provide insights into the relationship between COPD and AF and guide clinical practice. This systematic review and meta-analysis is designed to identify risk factors for new-onset AF in patients with COPD, and to formulate recommendations for preventing AF in COPD patients that will assist clinical decision making.

**Methods**. PubMed, Embase, Web of Science and Cochrane Library databases were searched for studies, which reported the results of potential risk factors for new-onset AF in COPD patients.

**Results**. Twenty studies involving 8,072,043 participants were included. Fifty factors were examined as potential risk factors for new-onset AF during COPD. Risk factors were grouped according to demographics, comorbid conditions, and COPD- and cardiovascular-related factors. In quantitative analysis, cardiovascular- and demographic-related factors with a greater than 50% increase in the odds of new-onset AF included age (over 65 years and over 75 years), acute care encounter, coronary artery disease, heart failure and congestive heart failure. Only one factor is related to the reduction of odds by more than 33.3%, which is black race (vs white). In qualitative analysis, the comparison of the risk factors was conducted between COPD-associated AF and non-COPD-associated AF. Cardiovascular-related factors for non-COPD-associated AF were also considered as risk factors for new-onset AF during COPD; however, the influence tended to be stronger during COPD. In addition, comorbid factors identified in non-COPD-associated AF were not associated with an increased risk of AF during COPD.

**Conclusions**. New-onset AF in COPD has significant demographic characteristics. Older age (over 65 years), males and white race are at higher risk of developing AF. COPD patients with a history of cardiovascular disease should be carefully monitored for new-onset of AF, and appropriate preventive measures should be implemented. Even patients with mild COPD are at high risk of new-onset AF. This study shows that risk factors for new-onset AF during COPD are mainly those associated with the

cardiovascular-related event and are not synonymous with comorbid factors for non-COPD-associated AF. The pathogenesis of COPD-associated AF may be predominantly related to the cardiac dysfunction caused by the chronic duration of COPD, which increases the risk of cardiovascular-related factors and further increases the risk of AF during COPD.

**PROSPERO registration number**. CRD42019137758.

## INTRODUCTION

Chronic obstructive pulmonary disease (COPD) is the fourth leading cause of morbidity and mortality worldwide (*GBD 2017 Mortality Collaborators, 2018*), and is predicted to rank third by 2030 (*Adeloye et al., 2015*). Atrial fibrillation (AF) often has a negative impact on patients with chronic disease, and is the most common arrhythmia in patients with COPD (*Kirchhof et al., 2018*). The co-existence of COPD and AF is associated with mutual exacerbation of the conditions, which complicates the management in clinical practice (*Gu et al., 2013*). AF in patients with COPD is associated with an accelerated decline in lung function, and a significant increase in cardiovascular accidents and long-term mortality (*Méndez-Bailón et al., 2017*). Reduced lung function also increases the risk of new-onset AF. In a prospective cohort study, the risk of new-onset AF was 1.8-times higher for forced expiratory volume in one second between 60%–80% (i.e., $60\% \leq FEV_1 < 80\%$) of predicted compared with $FEV_1 \geq 80\%$. In addition, the risk of AF related hospitalization was 1.3 times higher in $60\% \leq FEV_1 < 80\%$ of predicted and 1.8-times higher in $FEV_1 < 60\%$ of predicted compared with $FEV_1 \geq 80\%$ of predicted (*Buch et al., 2003*).

COPD is one of the main risk factors for AF. For patients with COPD, the high incidence of AF is combined with an alarming mortality rate. The incidence of AF in severe COPD patients is approximately four times higher than that in non-COPD patients (*Konecny et al., 2014*). The in-hospital mortality rate for patients with COPD-associated AF is 2.9%, whereas the corresponding rate for non-COPD-associated AF patients is 2.2% (*Méndez-Bailón et al., 2017*). Although the prevalence and short-term mortality rate of new-onset AF in COPD patients are high, the management of cardiovascular conditions in COPD patients is often ignored in current clinical practice. Elucidating the risk factors will provide an effective approach to screening high-risk COPD patients with new-onset AF and improving the clinical course and prognosis of COPD. A deeper understanding of the risk factors for new-onset AF in COPD patients will also provide insights into the relationship between COPD and AF and guide clinical practice. Based on the mutual promotion of deterioration between COPD and AF and the difference in primary disease settings, the risk factors for AF in the non-COPD setting do not adequately reflect the risk of AF in COPD patients. However, there is a paucity of information on the incidence of AF and insufficient evidence of risk factors in COPD patients from most studies.

Comprehensive integration of risk factors for AF in patients with COPD is important in bridging the knowledge gaps and providing information for the application of COPD guidelines. Therefore, this systematic review and meta-analysis was designed to identify risk factors for new-onset AF in patients with COPD, and to formulate recommendations for limiting AF in COPD patients that will assist clinical decision making.

## MATERIALS & METHODS

All methods of this systematic reviews and meta-analyses followed the Preferred Reporting Items for Systematic Reviews and Meta-Analyses (PRISMA) guidelines (*Wang et al., 2018a*). Details of the preregistered protocol for this study are available on International prospective register of systematic reviews (PROSPERO; CRD42019137758).

### Data sources and searches

The review authors searched for medical literature before December 2019. The search was conducted in four electronic databases including the Cochrane Library, PubMed, Embase, Web of Science (WOS), and the reference lists from review articles. The search strategy used a combination of MeSH, Emtree and text word, conducted with the following keywords: COPD or chronic obstructive pulmonary disease and AF or atrial fibrillation. The review articles with relevant topic were screened through keyword search, and their reference lists were further searched for eligible studies. The details of search strategies can be found in the Appendix S1. Search Strategy.

This meta-analysis included studies that met the following inclusion criteria:

1. Adult patients diagnosed with COPD (18 years of age or older). COPD was diagnosed according to the latest reference standards during the study, such as the GOLD criteria.
2. The studies reported the results of identified risk factors for new-onset AF in COPD patients. The adjusted odds ratio (OR), SE, and 95% confidence intervals (CI) for each candidate risk factor are reported as appropriate.
3. No publication date, status or language restrictions were applied.
   Exclusion criteria:
1. Secondary study, editorials, and animal experiments were excluded.
2. Studies that reported ORs with 95% CIs and *P* values showed inconsistency were excluded.
3. Studies that included only unadjusted OR and studies that involved new-onset AF in patients only after cardiothoracic surgery were excluded.

### Study selection

Two authors (T Shuai, C Zhang) independently reviewed the studies to be included in light of the titles, abstracts and keywords. If a study was found relevant to this topic, the full text would be further evaluated by at least two reviewers to assess whether it fulfills the selection criteria. In case of disagreement between the reviewers, the third reviewer (J Liu) would be consulted to resolve the disagreement. An inter-rater reliability analysis was conducted to assess the consistency of the study selection between two independent review authors. A study diagram was prepared for this selection to demonstrate the entire process of literature research and the selection of studies.

## Data extraction and quality assessment

The data was independently extracted by two review authors (Q Huang and H Xiong), and the resulting differences were resolved by a third reviewer (J Liu). The extracted data included: (1) citation information, participant characteristics, study types, and study settings; (2) definitions of COPD and AF; and (3) the adjusted OR, SE, and 95% CI for each candidate risk factor. Note that none of the studies from which data was extracted include more than one model where multiple variables were included, and any included study using empirical model building strategies (such as forward selection, stepwise approaches or backward elimination) may exclude variables from the adjusted models. If the data were insufficient, an email was sent to the authors to request the data. Note that all included studies provide complete information on the main outcomes.

Two review authors (J Liu and J Lu) independently applied the guidelines of the PRISMA statement to evaluate each involved study (*Liberati et al., 2009*). For observational studies such as case-control and cohort studies, the Newcastle-Ottawa Scale (NOS) was used (*Lo, Mertz & Loeb, 2014*) to assess the methodological quality and risk of bias (Tables S1 and S2). In case of any inconsistency, an agreement was reached through discussion between all authors.

## Data synthesis and risk factor analysis

Extracted data were analyzed using Stata SE 14.0 (Stata Corp; College Station, TX, USA). Considering the play of chance and some genuine variation in risk-factor effects, the pooled effect size was calculated by the random-effects method in this study. The heterogeneity of eligible studies was assessed by the Cochrane $Q$ test (substantive heterogeneity was indicated by $P < 0.05$) and the $I^2$ test (substantive heterogeneity was indicated by $I^2$ >50%). If substantive heterogeneity existed, sensitivity analysis was performed to analyze the potential sources of the heterogeneity. Egger's test was used for assessing the publication bias in case that there are over 10 included studies (*Egger et al., 1997*). The $\alpha$ value was set to 0.05.

Risk factors were grouped by factor type (demographic, comorbid condition, COPD-related factors and cardiovascular-related factors), and stratified by effect size, with specific consideration for factors associated with a greater than 50% increase or 33.3% decrease in odds of new-onset AF (the inverse of a 50% increase in odds corresponds to a 33.3% decrease in odds).

95% CIs were used to estimate the precision of the ORs. A wide CI indicates lower OR precision, whereas a narrow CI indicates higher OR precision. If a 95% CI spans 1.0, the increased or decreased odds does not reach statistical significance. Usually 95% CI excludes null value means $P < 0.05$. The 95% CI and $P$ value of potential risk factors involved in this study showed consistency.

### Quantitative analysis

Quantitative meta-analysis examined candidate risk factors identified in at least two studies (*Jackson et al., 2018*). The primary result was a pooled-adjusted OR for each risk factor using a general inverse variance method with random effects model. When the source

publication is unavailable, SEs were calculated using the following formula:

$$SE = \frac{ln(\text{Upper CI}) - ln(\text{Lower CI})}{3.92}$$

### *Qualitative analysis*

All included risk factors (pooled and un-pooled) were reported in at least one study. To assess whether increased risk factors for community-associated AF also increased the risk of COPD-associated AF, we compared the risk factors of new-onset AF in COPD with those of community-associated AF previously identified in *Chamberlain et al. (2011)* and *Schnabel et al. (2009)*.

## Patient and public involvement

This study is a meta-analysis using data from previously published studies, hence patients and the general public were not involved in this study.

## RESULTS

### Study selection and study characteristics

A flow chart of the study selection process prepared according to the PRISMA guidelines is presented in Fig. 1 (*Lo, Mertz & Loeb, 2014*). After reviewing the title and abstract, 150 articles were screened for full-text review. Of these, 130 articles failed to meet the inclusion criteria. Twenty studies were found to fulfill all the criteria and involved 8,072,043 participants. Inter-rater reliability was strong ($\kappa$, 0.80). Of the 20 included studies, 13 studies used retrospective cohort designs (*Alves Guimaraes et al., 2018*; *Chen & Liao, 2018a*; *Desai et al., 2019*; *Ganga et al., 2013*; *Genao et al., 2015*; *Hu & Lin, 2018*; *Koskela et al., 2014*; *Lainscak et al., 2009*; *Liao & Chen, 2017*; *Rusinowicz, Zielonka & Zycinska, 2017*; *Wang et al., 2018b*; *Warnier et al., 2010*; *Xiao et al., 2019*), three were prospective cohorts (*Hirayama et al., 2018*; *Short et al., 2012*; *Volchkova et al., 2015*), three were case-control studies (*Celli et al., 2010*; *Nadeem et al., 2015*; *Tomioka et al., 2019*), and one was a prospective nested case-control study (*Wilchesky et al., 2012*). All studies examined new-onset AF in patients with COPD. A summary of the characteristics of the included studies is shown in Table 1.

### Methodological quality and risk of bias

As shown in Table 1, the methodological quality of the observational studies was rated high, with 10 studies scoring nine of nine on the NOS (*Alves Guimaraes et al., 2018*; *Celli et al., 2010*; *Chen & Liao, 2018a*; *Desai et al., 2019*; *Genao et al., 2015*; *Short et al., 2012*; *Tomioka et al., 2019*; *Wang et al., 2018b*; *Warnier et al., 2010*; *Xiao et al., 2019*), six scoring eight of nine (*Hirayama et al., 2018*; *Hu & Lin, 2018*; *Koskela et al., 2014*; *Lainscak et al., 2009*; *Nadeem et al., 2015*; *Volchkova et al., 2015*), three scoring seven of nine (*Ganga et al., 2013*; *Liao & Chen, 2017*; *Wilchesky et al., 2012*), and one scoring six of nine (*Rusinowicz, Zielonka & Zycinska, 2017*). The overall risk of bias was rated low; however, the included studies used different disease definitions, which influenced the population selection. Summaries of the risk of bias are shown in Tables S1 and S2 (Appendix S2).

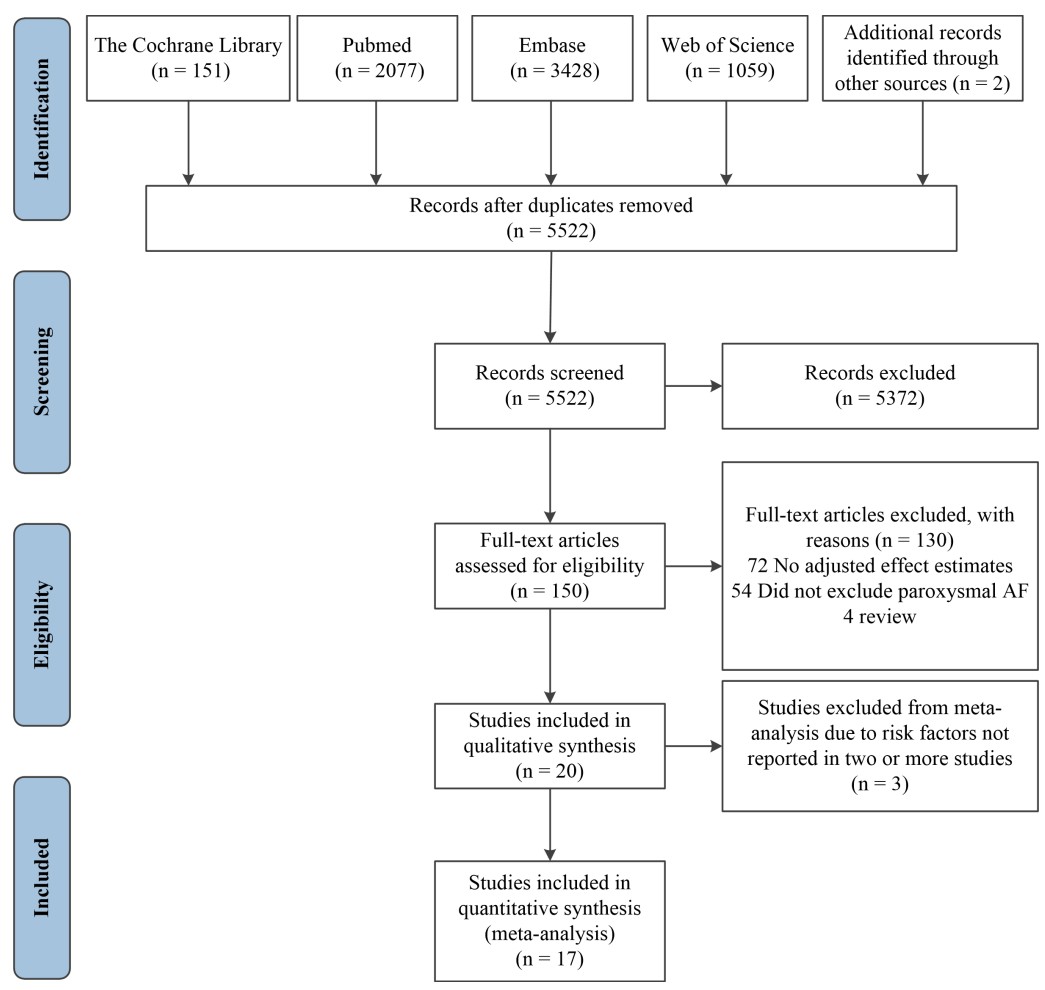

**Figure 1** PRISMA (preferred reporting items for systematic reviews and meta-analyses) flow diagram and exclusion criteria.

## Risk factor analysis
### Quantitative analysis
Among all risk factors, the 95% CI and *P* value of pooled OR showed consistency. Figure 2 shows the effect sizes of all 50 factors (un-pooled and pooled) examined as potential risk factors for new-onset AF in COPD patients. Twenty-five factors presented in at least two studies were included in the pooled meta-analysis. Seventeen pooled factors were associated with increased odds of new-onset AF: age (over 65 years and over 75 years), male, urban population, any acute care encounter, sepsis, renal failure, pneumonia, acute respiratory failure, invasive mechanical ventilation (IMV), noninvasive mechanical ventilation (NMV), ipratropium bromide use, short-acting $\beta$-agonist use, myocardial infarction (MI), coronary artery disease (CAD), heart failure (HF) and congestive heart failure (CHF). Of these, six factors were associated with a greater than 50% increase in odds: age (over 65 years and over 75 years), acute care encounter, CAD, HF and CHF.

Huang et al. (2020), *PeerJ*, DOI 10.7717/peerj.10376

**Table 1  Characteristics of included studies.**

| Author | Year | Country | Study Design | Study Period (Year) | Study Population | COPD with AF | Age (years) | COPD Definition | Identification of Atrial Fibrillation | Adjusted variables in logistic regression model | Newcastle-Ottawa Scale[a] | | |
|---|---|---|---|---|---|---|---|---|---|---|---|---|---|
| | | | | | | | | | | | Selection | Comparability | Outcome |
| Rupak Desai | 2019 | USA | Retrospective cohort | 2010-2014 | 6480799 | 4767401 | 75.8 | CCS code 127 | CCS code 106 | age, sex, race, admission type, median household income, length of stay, insurance payer, hospital bed size, ownership and location/teaching status of the hospital and all baseline comorbidities | **** | ** | *** |
| Xiaochun Xiao | 2019 | USA | Retrospective cohort | 2003-2014 | 1345270 | 244488 | 75.5 | ICD-9 codes 490.x, 491.x, 492.x, 494.x, and 496. x. | ICD-9 code 427 | age, sex, race, urban living, education[b], congestive heart failure, hypertension, diabetes mellitus, sepsis, acute respiratory failure, invasive mechanical ventilation, noninvasive mechanical ventilation, acute kidney injury and thromboembolism | **** | ** | *** |
| Tomoko Tomioka | 2019 | Japan | Case-control | 2010-2013 | 190 | 42 | 77.5 | GOLD | ECG review | age, brain natriuretic peptide level, the left atrial dimension and congestive heart failure | **** | ** | *** |
| Ya-Hui Wang | 2018 | China | Retrospective cohort | 2000-2011 | 12428 | 6219 | 71.2 | ICD-9 codes 491, 492, 496 | ICD-9 code 427.31 | age, sex, index year of AF, monthly income, hospital level, severe exacerbation of COPD in one year prior to index date (never, 1, or ≥2 times/year), medications for COPD, medications for hypertension, other medications and individual comorbidities | **** | ** | *** |
| Atsushi Hirayama | 2018 | USA | Prospective cohort | 2007-2012 | 944 | 400 | 77.3 | ICD-9-CM codes 491.21, 491.22, 491.8, 491.9, 492.8, 493.20, 493.21, 493.22, and 496 | ICD-9-CM code of 427.31 | age, sex, season and acute care | **** | * | *** |
| Chung-Yu Chen | 2018 | China | Retrospective cohort | 1997-2013 | 3528 | 882 | 67.0 | ICD-9-CM codes 490, 492, 496 | ICD-9-CM code 427.3 | age, sex, urban living, comorbidities, infection sites, and life-support treatments | **** | ** | *** |
| J.P. Alves Guimaraes | 2018 | Portugal | Retrospective cohort | 2010-2018 | 17573 | 2372 | 72.0 | ICD-9 codes 491, 492, 496 | ICD-9 code 427.31 | age, sex, body mass index, and comorbidities | **** | ** | *** |
| Wei-Syun Hu | 2018 | China | Retrospective cohort | 1995 | 51835 | 1492 | 72.5 | ICD-9-CM codes 491, 492, 496 | ICD-9-CM code of 427.31 | age (<64, 65-74, and >75), sex, hyperlipidemia, chronic kidney disease, vascular disease and other comorbidities | **** | ** | ** |
| Kuang Ming Liao | 2017 | China | Retrospective cohort | 1997-2013 | 6208 | 1547 | 65.3 | ICD-9-CM codes 490- 492, 496 | ICD-9-CM code 427.3 | age, sex, race, urban living, comorbidities and infection sites | *** | * | *** |

Huang et al. (2020), *PeerJ*, DOI 10.7717/peerj.10376

**Table 1** (*continued*)

| Author | Year | Country | Study Design | Study Period (Year) | Study Population | COPD with AF | Age (years) | COPD Definition | Identification of Atrial Fibrillation | Adjusted variables in logistic regression model | Newcastle-Ottawa Scale[a] | | |
|---|---|---|---|---|---|---|---|---|---|---|---|---|---|
| | | | | | | | | | | | Selection | Comparability | Outcome |
| Tomasz Rusinowicz | 2017 | Poland | Retrospective cohort | 2004-2016 | 152 | 46 | 75.0 | GOLD | 24-h ECG review | age, sex, race, education, smoking habits and pack-years, income, body mass index, comorbid conditions and medication use | ** | * | *** |
| Rashid Nadeem | 2015 | USA | Case-control | 2008-2012 | 312 | 68 | 73.4 | GOLD | ECG review | age, sex, ischemic cerebrovascular accident, diabetes mellitus, hypertension, peripheral vascular disease, hyperlipidemia, and congestive cardiac failure. | **** | * | *** |
| E.A.V. Volchkova | 2015 | Russian | Prospective cohort | 2009-2011 | 229 | 70 | 67.2 | GOLD | ECG review | age, sex, body mass index, left ventricle end-diastolic volume, left atrial enlargement, left ventricular hypertrophy and comorbidities | *** | ** | *** |
| Liza Genao | 2015 | USA | Retrospective cohort | 2005-2011 | 52741 | 9971 | 77.0 | ICD-9-CM codes 491.x, 492.x, or 496.x | ICD-9-CM 420–429 | Age, sex, acute care, recurrent AE-COPD | **** | ** | *** |
| Jukka Koskela | 2014 | Finland | Retrospective cohort | 1995-2006 | 505 | 31 | 64.0 | ICD-10 | ICD-10 | age, sex, the health related quality of life[c] and comorbidities | *** | ** | *** |
| Harsha V. Ganga | 2013 | USA | Retrospective cohort | 2006 | 416 | 44 | 77.6 | GOLD | ICD 9 codes 427.3, 427.31 and 427.32 | obstructive sleep apnea, heart failure, hypertension, chronic kidney disease, diabetes, hyperlipidemia and valve disorders | **** | * | ** |
| Machelle Wilchesky | 2012 | Canada | Prospective nested case-control | 1990-2002 | 76661 | 2339 | 80.3 | GOLD | ECG review | medication use, COPD disease severity, cardiovascular disease, and other comorbidities | ** | ** | *** |
| PM Short | 2012 | UK | Prospective cohort | 2009-2012 | 1343 | 155 | 72.0 | GOLD | ICD 9 codes 427 | age, urban living, prior beta blocker use, MRC dyspnoea score and comorbidities that associated with AF | **** | ** | *** |
| Bartolome Celli | 2010 | Germany | Case-control | 2008 | 19545 | 306 | 65.0 | GOLD | ECG review | age, sex, urban living, tiotropium use, comorbidities and smoking habits and pack-years | **** | ** | *** |
| Miriam J Warnier | 2010 | Netherlands | Retrospective cohort | 2009 | 404 | 22 | 72.9 | GOLD | 12-lead ECG review | age, sex, diabetes mellitus, myocardial infarction, and antiarrhythmic QT prolonging drugs use. | **** | ** | *** |
| Mitja Lainscak | 2009 | Germany | Retrospective cohort | 2002-2007 | 960 | 288 | 71.0 | ICD-10 | ICD-10 | age, sex, tobacco consumption and comorbidities | *** | ** | *** |

**Notes.**

Abbreviations: COPD, Chronic obstructive pulmonary disease; AF, Atrial fibrillation; GOLD, The Global Initiative for Chronic Obstructive Lung Disease; ICD-9-CM codes, the International Classification of Diseases-Ninth Revision-Clinical Modification; CCS codes, the Clinical Classifications Software; ECG, electrocardiogram; NA, not available.

[a]The Newcastle-Ottawa Scale (NOS) assesses the quality of case-control and cohort studies based on categories of selection, comparability, and outcome (or exposure for case-control). NOS uses a star-based system where more stars represent higher quality within a specific category. Studies are awarded a maximum of four stars (****) for selection, two stars (**) for comparability, and three stars (***) for outcome.

[b]"Education" refers to the completion of high school education.

[c]The health related quality of life (HRQoL) was assessed using the self-administered Airways Questionnaire 20 (AQ20). Low HRQoL is defined as AQ20 summary score ≥14.

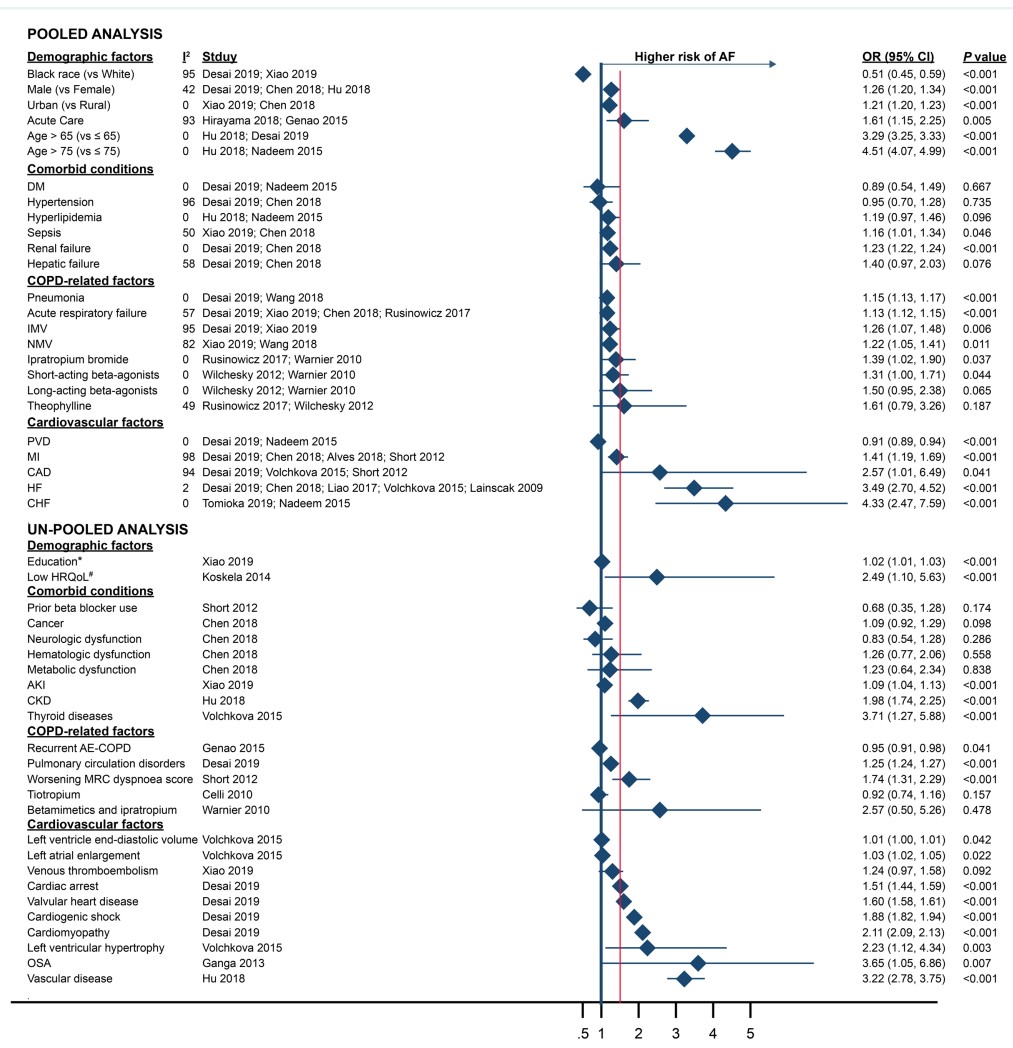

**Figure 2 Risk factors for AF in COPD patients stratified by factor type and pooled versus un-pooled analysis.** Filled diamonds show the pooled-adjusted odds ratios (OR) for risk factors from the meta-analysis and the adjusted OR for un-pooled risk factors. Error bars denote 95% CIs. Abbreviations: COPD, Chronic obstructive pulmonary disease; AF, Atrial fibrillation; OR, Odds ratio; 95% CI, Confidence intervals; IMV, Invasive mechanical ventilation; NMV, Noninvasive mechanical ventilation; MI, Myocardial infarction; CAD, Coronary artery disease; HF, Heart failure; CHF, Congestive heart failure; PVD, Peripheral vascular disease; DM, Diabetes mellitus; HRQoL, the Health related quality of life; AKI, Acute kidney injury; CKD, Chronic kidney disease; OSA, Obstructive sleep apnea.* Below high school degree (vs high school degree or above). # The HRQoL was assessed using the self-administered Airways Questionnaire 20 (AQ20). Low HRQoL is defined as AQ20 summary score ≥14.

Two pooled factors were associated with decreased odds of new-onset AF in the meta-analysis: black ethnicity (compared with white), and peripheral vascular disease (PVD). Only one factor is related to the reduction of odds by more than 33.3%, which is black race (vs white). Six factors were not associated with new-onset AF after the pooled analysis: hypertension, hyperlipidemia, diabetes mellitus (DM), hepatic failure, long-acting beta-agonist use and theophylline use.

*Heterogeneity*

Heterogeneity analysis showed that $I^2$ was greater than 50% in nine pooled risk factors, seven of which were included the study by *Desai et al. (2019)*. The direction of the effects in individual study ORs from five of these seven pooled factors included in the study by *Desai et al. (2019)* was concordant (black ethnicity, hepatic failure, IMV, MI and CAD). The direction of the effect estimates in individual ORs from the other two pooled risk factors (hypertension and acute respiratory failure) was discordant. The other two pooled factors with high heterogeneity were not included in the study by *Desai et al. (2019)* (acute care and NMV) and both included only two studies that reported the risk factor and had the same direction of effect for individual study ORs.

Sensitivity analysis of acute respiratory failure, MI and CAD showed that the high heterogeneity was derived from the study by *Desai et al. (2019)*. The results of sensitivity analysis are shown in Figs. S1– S3 (Appedix S3. Sensitivity analysis). When the study by *Desai et al. (2019)* was excluded from the analysis, $I^2$ decreased from 57% to 0% and the OR decreased from 1.13 (95% CI [1.12–1.15]) to 1.09 (95% CI [1.06–1.12]) for acute respiratory failure, $I^2$ decreased from 98% to 63% and the OR increased from 1.41 (95% CI [1.19–1.69]) to 1.55 (95% CI [1.29–1.85]) for MI and $I^2$ decreased from 94% to 35% and the OR increased from 2.57 (95% CI [1.01–6.49]) to 3.88 (95% CI [2.43–6.21]) for CAD.

## Qualitative analysis

All of the un-pooled ORs showed the 95% CI and *P* value with consistency. To qualitatively compare the risk factors of COPD-associated AF and community-associated AF risk factors, we included 25 additional factors that were evaluated in only one study. Eleven un-pooled factors were associated with a greater than 50% increase in odds of new-onset AF, including lower health-related quality of life (HRQoL; HRQoL was assessed using the self-administered Airways Questionnaire 20 (AQ20). Low HRQoL is defined as AQ20 summary score ≥ 14), chronic kidney disease (CKD), vascular disease, thyroid diseases, worsening MRC dyspnea score, history of cardiac arrest, valvular heart disease, history of cardiogenic shock, cardiomyopathy, left ventricular hypertrophy, and obstructive sleep apnea (OSA). No factor was associated with a greater than 33.3% decrease in the odds. Eight factors were not associated with new-onset AF in terms of odds, including cancer, neurologic dysfunction, hematologic dysfunction, metabolic dysfunction, prior beta-blocker use, tiotropium use, betamimetics and ipratropium use and venous thromboembolism.

Of the 11 community-associated factors identified by *Schnabel et al. (2009)* and the 20 community-associated factors identified by *Chamberlain et al. (2011)*, 10 factors had similar definitions compared with COPD-related factors evaluated in the current study, which allowed a direct comparison. A summary of the comparison of the effect sizes of risk factors for COPD-associated versus community-associated AF is shown in Table 2. In general, comorbid conditions associated with increased risk for community-associated AF (e.g., DM, prior beta-blocker use, and hypertension) were not associated with higher risk of new-onset AF in COPD patients. Cardiovascular-related factors for community-associated

**Table 2  Comparison of risk factors for COPD-associated AF and Non-COPD-associated AF.**

| Risk Factors | COPD-Associated AF[a] | Effect Size | |
| --- | --- | --- | --- |
| | | **Non-COPD-Associated AF** | |
| | | Schnabel et al.[b] | Chamberlain et al.[b] |
| Black race (vs White) | 0.51 (0.45–0.59) | 0.53 (0.44–0.63) | 0.52 (0.43–0.62) |
| Male (vs Female) | 1.26 (1.20–1.34) | 1.90 (1.58–2.29) | 1.92 (1.60-2.30) |
| Age >65 (vs ≤ 65) | 3.29 (3.25–3.33) | 2.28 (2.08–2.49) | NA |
| DM | 0.89 (0.54–1.49) | 1.10 (0.87–1.38) | 1.87 (1.51–2.32) |
| Prior beta blocker use | 0.68 (0.35–1.28) | 1.80 (1.48–2.18) | 2.55 (2.13–3.04) |
| Hypertension | 0.95 (0.70–1.28) | NA | 2.16 (1.67–2.79) |
| MI | 1.41 (1.19–1.69) | 1.34 (1.02–2.03) | NA |
| CAD | 2.57 (1.01–6.49) | NA | 2.21 (1.71-2.84) |
| HF | 3.49 (2.70–4.52) | 3.20 (1.99–5.16) | 3.03 (2.32–3.95) |
| Left ventricular hypertrophy | 2.23 (1.12–4.34) | 1.36 (1.03–1.80) | NA |

Notes.

Abbreviations: COPD, Chronic obstructive pulmonary disease; AF, Atrial fibrillation; DM, Diabetes mellitus; MI, Myocardial infarction; CAD, Coronary artery disease; HF, Heart failure; NA, not available.

[a]Odds ratio (95% CI)

[b]Hazard ratio (95% CI)

Shown are effect sizes for factors evaluated in current study (COPD-associated AF) and by *Schnabel et al. (2009)* and *Chamberlain et al. (2011)* (non-COPD-associated AF) that are similarly defined.

AF were also identified as risk factors for new-onset AF in COPD patients, and the effective estimates tended to be stronger in COPD patients.

## DISCUSSION

In this, study, we conducted a systematic review and meta-analysis to characterize and evaluate risk factors for new-onset AF in COPD patients. Fifty potential risk factors were grouped by factor type. Quantitative analysis demonstrated that risk factors for new-onset AF in COPD patients are mainly those associated with cardiovascular-related events and demographic factors (e.g., age over 65, age over 75, acute care encounter, CAD, HF and CHF). Although COPD-related factors can also trigger new-onset AF in COPD patients, the risk effect is moderate. In this study, we further explored the differences between the risk factors for COPD-associated AF and the identified risk factors for non-COPD-associated AF. Although cardiovascular-related factors for non-COPD-associated AF are also considered to be risk factors for new-onset AF in COPD patients, the influences are stronger in COPD patients. Interestingly, comorbid factors (DM, prior beta-blocker use, hypertension, PVD and hepatic failure) identified in non-COPD-associated AF are not associated with an increased risk of new-onset AF in COPD patients. This highlights the importance of distinguishing risk factors for AF in patients with COPD and the potential need for different preventive interventions.

New-onset AF in COPD patients has significant demographic characteristics. Older age (over 65 years and over 75 years), male and whites are at higher risk of morbidity. Among all the demographic factors, older age is the dominant factor (over 65 years: OR = 3.29; over 75 years: OR = 4.51). Both COPD and AF have age distribution characteristics, and

the prevalence increases with age. In the study conducted by Méndez-Bailón et al., they found that the prevalence in AF patients with COPD also showed a increasing tendency with age. Their results indicate that the overall prevalence of COPD was approximately 17% in AF patients, increasing to more than 40% in individuals aged >75 years. (*Méndez-Bailón et al., 2017*). This information highlights demographic characteristics that can be used by clinicians to pre-screen high-risk populations.

The most commonly identified risk factors for new-onset AF in COPD patients are associated with cardiovascular events. Although these factors are also considered to be risk factors for non-COPD-related AF, the risk of cardiovascular events in patients with COPD is often amplified compared to that in non-COPD patients. As a chronic disease, COPD is independently associated with cardiovascular risk (*Sidney et al., 2005*). The adverse effects of COPD on the heart are multifaceted. This condition not only exacerbates organic heart disease and directly induces AF, but also induces hypoxia, hypercapnia (*Ferraro et al., 2019*), electrolyte disturbances and increased blood viscosity, which exacerbate the cardiac burden and oxygen consumption, and may therefore lead to deterioration of cardiac function in the presence of concomitant cardiovascular disease (*Feary et al., 2010*; *Fuster et al., 2006*). Furthermore, sympathetic nerve overactivity and the aggravation of COPD-associated inflammation, especially in acute exacerbation of COPD, can lead to increased heart rate and cardiac events (*Andreas et al., 2005*; *Lopez & House-Fancher, 2005*). Due to the mutual exacerbation of COPD and cardiovascular disease, the risk of cardiovascular-related factors is more notable in patients with COPD.

In the chronic process, patients with chronic COPD have a higher risk of cardiac abnormalities (including HF, CAD and MI) than non-COPD patients. These organic cardiopathies contribute to the occurrence of AF. The results of this study demonstrate a potential association between cardiovascular-related history and new-onset AF in patients with COPD. However, in clinical practice, clinicians often ignore the history of heart disease in patients with COPD (*Konecny et al., 2014*). Attention to the cardiac history of COPD patients can help with early management and clinical decision-making regarding the necessary interventions, thereby effectively reducing the risk of new-onset AF in COPD patients.

Although the risk effect of COPD-related factors is not as strong as cardiovascular and demographic factors, the impact on COPD-associated AF remains substantial. Moreover, we found that slight decreases in the pulmonary function of COPD patients were associated with increased risk of AF; therefore, this situation requires careful monitoring and prevention. As a remedy for NMV failure, IMV usually indicates a further deterioration of pulmonary function in COPD patients. We found that not only IMV, but also NMV, risk triggering AF. This finding is consistent with the study by *Buch et al. (2003)*, which showed that the risk of triggering new-onset AF was 1.8-times higher for $60\% \leq FEV_1 < 80\%$ of predicted compared with $FEV_1 \geq 80\%$.

In addition, we found that COPD-related factors have the potential to trigger AF (e.g., pneumonia, acute respiratory failure). In accordance with our findings, previous studies have shown that new-onset AF in COPD patients can be caused by COPD-related factors, such as the inflammatory state of COPD and the use of beta-agonists. In animal models,
streptococcus-induced pneumonia has been shown to cause myocardial micro-abscesses, which can develop into fibrotic lesions prone to arrhythmia formation (*Ganga et al., 2013*). The formation of arrhythmic lesions may explain our findings of increased risk of AF in patients with pneumonia.

Comorbid conditions are major factors that can be used to predict the risk of community-associated AF. However, our findings suggest that the risk effects of comorbid conditions are low compared with the effects of cardiovascular-related or COPD-related factors. This may reflect the greater severity of the disease (e.g., HF, CAD and MI) in COPD patients or fewer competing risk factors, resulting in a lower proportion of comorbid factors attributed to new-onset AF in patients with COPD (*Chamberlain et al., 2011*; *Vestbo et al., 2013*). Compared with other comorbid conditions, COPD itself is the greater risk for triggering AF.

This study of risk factors contributed to our understanding of the pathogenesis of AF in COPD patients. AF is thought to occur in two steps that involve the production of arrhythmogenic substrates (*Liu et al., 2017*), followed by the triggering event (*Aldhoon et al., 2010*). In this analysis, we identified chronic factors that are known to potentiate the formation of atrial fibrosis following long-term exposure (e.g., CHF and CAD), which deteriorate further in the setting of COPD. We also identified several COPD-related factors that may trigger AF (e.g., pneumonia, acute respiratory failure, IMV and NMV). However, the risk effect of cardiovascular factors was found to be greater than that of COPD-related factors. This suggests that new-onset AF in patients with COPD is caused mainly by the deterioration in cardiac function during chronic COPD, with the acute deterioration of COPD itself affecting the occurrence of AF.

There are several limitations in the current systematic review and meta-analysis. Firstly, there was substantial heterogeneity in the results of some risk factors. In order to respond to this limitation, we then performed a sensitivity analysis to explore the potential sources of heterogeneity. We identified the study by *Desai et al. (2019)* as the major source of heterogeneity in three risk factors, including acute respiratory failure ($I^2$ decreased from 57% to 0%), MI ($I^2$ decreased from 98% to 63%) and CAD ($I^2$ decreased from 94% to 35%). The analysis of heterogeneity is limited due to the insufficient number of studies reporting risk factors. In addition, other potential sources of heterogeneity in our analysis of the results of baseline characteristics are as follows: (1) The designs of the included studies were heterogeneous. Compared with meta-analyses including randomized controlled trials, those including observational studies have greater potential bias. (2) The studies included patients with diseases defined using different strategies. Although all strategies are reasonable, different criteria may lead to variations in clinicians' decision-making, which may increase the heterogeneity of population selection.

Secondly, interpretation of the observed ORs depends on all necessary confounders being included, without any colliders or intermediate variables. However, the adjusted ORs reported in the included studies cannot eliminate residual confounding caused by unknown, missed and inaccurate confounding factors, and there is a lack of discussion about the causality of the variables. Therefore, limited by the existing methods, it is difficult to analyze the causal associations of 50 variables without additional original research. Furthermore,

we also suggest more original research to use the causal modelling approaches such as directed acyclic graphs to distinguish between confounders, colliders, and intermediate variables when considering causal risk factors. Thirdly, due to different research purposes, the included studies only reported the results of some potential risk factors related to the research purpose, while other variables were only mentioned as an adjusted variable, and did not report the adjusted ORs and 95% CIs. Thus, the number of risk factors included in the meta-analysis was less than the number of adjusted variables included in the adjusted models. This suggests that more original studies are needed to report more comprehensive results for further analysis and evaluation of specific risk factors. In addition, because no risk factor was reported in 10 or more original studies; Egger's test and funnel plots were not included in our analysis. Further high-quality original studies can also help to limit the potential publication bias.

## CONCLUSIONS

This systematic review and meta-analysis reveals the demographic characteristics of patients with new-onset AF in COPD patients. Older age (over 65 years and over 75 years), males and whites are at higher risk of developing AF. The dominant factors are cardiovascular (CAD, HF and CHF), and may be amplified in the context of COPD. COPD patients with a history of cardiovascular disease should be carefully monitored for new-onset AF, and appropriate preventive measure should be implemented. Even for patients with mild COPD, clinicians should not relax their vigilance in monitoring patients for new-onset AF. The pathogenesis of AF in COPD patients may be primarily related to cardiac dysfunction caused by the chronic duration of COPD, which increases the risk effect of cardiovascular-related factors and further increases the risk of AF in patients with COPD. This highlights the importance of identifying risk factors for predicting COPD-associated AF and the potential need for different preventive interventions.

## ACKNOWLEDGEMENTS

The authors gratefully acknowledge the support of the First Clinical Hospital of Lanzhou University, the first clinical medical college of Lanzhou University, Evidence-based Medicine Center of Lanzhou University, and all the authors who participated in this study.

### Funding

The authors received no funding for this work.

### Competing Interests

The authors declare there are no competing interests.

## Author Contributions

- Qiangru Huang conceived and designed the experiments, performed the experiments, prepared figures and/or tables, authored or reviewed drafts of the paper, and approved the final draft.
- Huaiyu Xiong conceived and designed the experiments, performed the experiments, authored or reviewed drafts of the paper, and approved the final draft.
- Tiankui Shuai performed the experiments, authored or reviewed drafts of the paper, and approved the final draft.
- Meng Zhang, Chuchu Zhang and Yalei Wang analyzed the data, authored or reviewed drafts of the paper, and approved the final draft.
- Lei Zhu and Jiaju Lu performed the experiments, prepared figures and/or tables, authored or reviewed drafts of the paper, and approved the final draft.
- Jian Liu conceived and designed the experiments, performed the experiments, analyzed the data, authored or reviewed drafts of the paper, and approved the final draft.

## Data Availability

The raw data are available in Tables 1 and 2.

## Supplemental Information

Supplemental information for this article can be found online at http://dx.doi.org/10.7717/peerj.10376#supplemental-information.

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
