# Peer review of "Risk factors for new-onset atrial fibrillation in patients with chronic obstructive pulmonary disease: a systematic review and meta-analysis"

_PeerJ, doi:10.7717/peerj.10376_

## Round 0.1 · original submission · Major Revisions

In light of the reviewers' comments, I think your manuscript has scientific validity. However, there are some issues that you must address in a revised version of the text.

Reviewer 1 ·

Basic reporting

This article suggested the Risk factors for new-onset atrial fibrillation in patients with chronic obstructive pulmonary disease. In general, this article bears significance and innovation. I recommend revision for this article. The following comments are provided for the authors' consideration before accept.

Experimental design

The Experimental design is good, as least in part.

Validity of the findings

The findings are helpful for the cardiologists and respiratory specialists.

Additional comments

This article suggested the Risk factors for new-onset atrial fibrillation in patients with chronic obstructive pulmonary disease. In general, this article bears significance and innovation. I recommend revision for this article. The following comments are provided for the authors' consideration before accept.

Major consideration
1. The search strategy is terrible and the authors shall used a combination of MeSH, Emtree and text word, and give a detailed search strategy including exp time, literature number of each database.
e.g. PubMed: please use methods of MeSH and text word.
Embase: Emtree
as well as Cochrane and ClinicalTrials.gov
2. The type setting is terrible and must be in accordance with the style of the journal.
3. The assessment of risk of bias is missing.
4. The subgroup analysis is missing. E.g. Retrospective cohort AND case-control; Asian OR Euro-America.
5. The funnel plot is missing.
6. The keywords is missing.

Minor consideration
1. The language needs revision by a native speaker, e.g. American Journal Experts.
2. Please give reference for Quantitative Analysis.
3. Please provide a PRISMA Checklist.
4. Inclusion/exclusion criteria shall be in detail.
5. Please cite the following literature in appropriate sites throughout the manuscript for a better understanding of background and discussion, and marked them in red clearly in the reference. PMID:
COPD: 30659954
atrial fibrillation: 28400980

·

Basic reporting

Clear and unambiguous English, good structure and sufficient literarure

Experimental design

Research question well defined. The data are well-presented, the study is methodologically right and the results are very interesting and fill an identified knowledge gap.

Validity of the findings

Important results, conclusions well-stated.

Additional comments

This study was designed to identify risk factors for new-onset AF in patients with COPD, and to propose recommendations for preventing AF in COPD patients and to assist clinical decision making.


The data are well-presented, the study is methodologically right and the results are very interesting. The study reports that cardiovascular and demographic factors are related to new-onset AF in patients with COPD. Despite the fact that cardiovascular related factors of non-COPD-associated AF are also considered as risk factors for new-onset AF during COPD, influences are stronger in COPD patients. Interestingly, co-morbid factors (diabetes mellitus, prior beta blocker use, hypertension, peripheral vascular disease and hepatic failure) identified in non COPD-associated AF are not associated with an increased risk of new-onset AF during COPD. This highlights the importance of distinguishing risk factors for AF in patients with COPD and the potential need for different preventive interventions.


The study deserves to be published.

Minor points

1) In my opinion I think that the title of the study should not include the phrase “based on 8,072,043 participants”. The title should be “Risk factors for new-onset atrial fibrillation in patients with chronic obstructive pulmonary disease: a systematic review and meta-analysis”.
2) In page 14, line 276, “expiratory”, instead of “expiatory”.
3) In page 14, line 281, “streptococcus-induced pneumonia”, instead of “streptococcus pneumonia-induced pneumonia”.

·

Basic reporting

Some of the wording is awkward and needs attention. For example, “In qualitative analysis, comparison of the risk factors for COPD-associated and identified non-COPD-associated AF.” (Lines 56–58). Careful proofreading by a native English speaker should identify and correct these issues. There are also various typos (e.g. “andthe” in Table 1) that could also be identified and corrected in this way.

Note a typo in each of Figures S1, S2, S3 (“ommited”).

There are sometimes missing spaces, e.g. the typo above, Line 138’s “StataSE14.0”, Table 2 and Figure 2’s “Black race(vs White)”, etc.

Sometimes the manuscript talks about “gender” (the social construct) and sometimes “sex” (the biological status), e.g. Table 1 includes both.

Rather than stating databases “including” those listed on Lines 106–107, an exhaustive list of databases should be provided at this point in the text.

There seems to be some potential confusion when describing the database searches (Lines 105–109) and review article checks (Line 107) which do not involve keyword searches (at least not in the same way).

Experimental design

When multiple adjusted models were identified, assuming this occurred during data extraction, how was the “adjusted” OR (Line 114) determined? Was this the most adjusted model and how were models that adjusted for variables potentially on the causal pathway as sensitivity analyses, for example, handled as part of this process for identifying the data to be extracted? Note that the most appropriate causal model could differ between risk factors also (a variable that is a confounder or competing exposure for one association could be a mediator or collider for another association).

While I commend the authors for contacting authors of studies where they were not able to extract necessary information (Lines 129–130), how often did this situation arise and how often did authors provide the requested data?

It should be made clearer what variables were examined for inter-rater reliability (Line 166, is this simply for studies being selected yes/no?) and exactly when this was done (pre-discussion?)

I would have preferred a more theory-driven/principled approach for deciding between using a fixed or a random (not “randomized”, Line 140) effects model as the interpretation of these is different. It would not, in particular, make sense to me to use both approaches for different exposures from the same study unless there was a clear and well-explained mechanism that would justify treating the factors differently. While statistical tests for and measures of heterogeneity are appealing in many ways, I would suggest a more principled approach be taken here (Lines 138–143) and suggest looking at sections 9.5 and 9.6 from the Cochrane Handbook. Can the authors justify their empirical strategy here?

Validity of the findings

The phrase “strongest risk factors” (e.g. Line 54) is problematic for me as the meaning of “strongest” here depends on the reader’s interpretation (for example, this could refer to the magnitude of the effect for some particular comparison involving that factor or the statistical significance of an association involving that factor). Neither of these can fully inform the clinical interpretation of the potential risk factors (for some variables at least) which will also potentially include modifiability and ease of assessment for screening variables. The use of a 50% increase in odds for identifying important factors is also potentially problematic for at least some variables as this depends crucially on the categories used for categorical variables (including the level used as the reference) and the units used for continuous variables (and note that this is not a 50% increase in “likelihood” as stated on Line 147, c.f. Line 186). For the former, an OR of 1.8 for one non-reference level and an OR of 1.1 for another non-reference level would, for equal numbers in each non-reference level, average out as less than 1.50 for a pooled non-reference level despite one category meeting this criterion (compared to the reference and also compared to a pooled category involving the lower risk non-reference level) showing also the importance of how dichotomous variables are determined. Did any of the studies report on results using three (or more) categories and if so how were these addressed (e.g. ethnicity as Black, White, and Hispanic or Other)? Or were all binary variables included using the same two labels and definitions throughout (this might also inform the choice of fixed versus random effects models)? Similarly, for the latter, an OR of 1.042 for one unit of a continuous variable would represent an OR of 1.51 for a 10 unit increase or a comparison between categories with means 10 units apart, but less than 50% higher odds for any smaller increment. Categorising continuous variables, such as age, cannot entirely avoid this problem as the number and widths of categories can potentially make the reported OR either greater or less than 1.5. I appreciate the desire to identify substantial risk factors, but the approach used here seems fraught to me outside of clear dichotomous variables (e.g. presence/absence of a well-defined co-morbidity). Was this always the case?

At the same time, this approach to identifying important factors ignores the precision/lack of precision communicated by the widths of the CIs. An OR of 2 with 95% CI limits of 0.5 and 8 would meet the criterion of OR > 1.5 despite the large uncertainty. Restricting these factors to only those statistically significant would then ignore an OR of 4 with 95% CI limits of 0.99 and 16.16 despite most of the 95% CI being above the criterion value (the interpretation here would be facilitated by credible intervals instead).

At the same time, the interpretation of any risk factor depends crucially on other variables included in the model. The clearest interpretation of the observed ORs depends on all necessary confounders being included but no colliders or intermediate variables, with competing exposures included to enhance precision. I appreciate that you cannot control the models used in each of the included studies, but I do feel that more consideration of these points (and perhaps some sensitivity analyses informed by them) is needed along with some incorporation of this as a limitation in the discussion. I wonder if an “ideal” DAG (directed acyclic graph) might not be useful for readers to appreciate how you see the causal associations that inform the interpretation of the published results, and such a model would seem likely to be of great use to researchers in this area.

Additional comments

My apologies for my slow review in these trying times.

---

## Round 0.2 · Minor Revisions

The manuscript still needs some minor modifications. Please refer to the comments from Reviewer 3.

Reviewer 1 ·

Basic reporting

The authors have addressed questions successfully.

Experimental design

The authors have addressed questions successfully.

Validity of the findings

The authors have addressed questions successfully.

Additional comments

The authors have addressed questions successfully.

·

Basic reporting

Clear and unambiguous English, good structure and sufficient literarure.

Experimental design

Research question well defined. The data are well-presented, the study is methodologically right and the results are very interesting and fill an identified knowledge gap.

Validity of the findings

Important results, conclusions well-stated.This study investigated the differences between the risk factors of COPD-associated AF and the risk factors of non-COPD-associated AF. The authors reported that demographic characteristics have a significant impact on new-onset AF during COPD. Older age (over 65 years of age), men and whites are at higher risk of developing AF. Cardiovascular related factors (CAD, HF and CHF) are dominant factors as well, in this setting. Of note, co-morbid factors (DM, prior beta blocker use, hypertension, PVD and hepatic failure) identified in non-COPD-associated AF are not associated with increased risk of new-onset AF during COPD. Therefore, it seems to be of outmost importance to distinguish risk factors for AF in patients with COPD and to take appropriate preventive measures, especially in patients with a history of cardiovascular disease.

Additional comments

The authors have carefully revised the manuscript and performed the necessary changes according to the reviewers’ comments. The data are presented in detail, the study is methodologically right and the results are very interesting.

·

Basic reporting

Some specific/minor comments are below:

Line 49: I think you mean “included” rather than “including” here and don’t need the immediately preceding comma.

Line 146: Thank you for explaining that this was not necessary. I think some readers would also appreciate this information as a note around here.

Line 167: “wide” would be a clearer description than “large” when it comes to CIs (although, all other things being equal, larger ORs will also have wider CIs).

Lines 168–169: While the 95% CI includes/does not include the null value == non-/statistical significance rule works for linear models, it doesn’t quite work for binary outcomes as the standard error differs for the hypothesis test and the CI, so it is possible to generate data where the 95% (as you note on Line 167) CI for the OR is strictly above or below 1 and it is not statistically significant, and where the CI includes 1 but is statistically significant at the 0.05 level (as you note on Line 162).

Line 168: Similarly, “narrow” rather than “small”.

Lines 349–351: I think you mean that a limitation was that there was substantial heterogeneity. In response to this limitation, you then performed these additional analyses.

Line 365: Rather than “required” (which is past tense), do you mean “require”, or perhaps “recommend” or “suggest”.

Line 366: Rather than “the directed acyclic graph”, perhaps “causal modelling approaches such as directed acyclic graphs” as it is the process that matters rather than the specific technique used to achieve this.

Lines 366–367: Rather than “to distinguish the confider, collider, and mediator in risk factors”, perhaps “to distinguish between confounders, colliders, and mediators when considering causal risk factors”?

Table 1: Your column “Study Inception” shows the year the study started (its inception) and the year it finished (its conclusion), perhaps “Study period” would be more correct? Rather than “Adjustment” do you mean this to be a full set of “Independent variables”? For Miriam Warnier’s study, you list “sex, age,…” in the variables when you usually do this in the opposite order.

Figure 2: There seems to be a space in “bromide” (i.e. “brom ide”).

PRISMA checklist: You are missing a space after “page” for the first two items. You also vary capitalisation (e.g. “page” sometimes and others “Page”) in this column.

Experimental design

See below.

Validity of the findings

See below.

Additional comments

Thank you for your constructive and informative responses and the edits to your manuscript. I feel that your work is much improved, although I do still have some questions remaining.

Sorry, but it’s still not absolutely clear to me how you determined the “adjusted model”. Did none of the articles from which data was extracted include more than one model where multiple variables were included? If so, this fact could be added around Line 145. The same point would apply to any article using forward selection, stepwise, backward elimination, LR test-based, or IC-based approaches where variables would not appear in the “adjusted” model if they were excluded because of these empirical model building strategies. The same reassurance, or warning, could be added around Line 125, 130, or 145. I note that in Table 1, for the “Adjustment” variables for Harsha Ganga’s study, you say “clinical variables that were significantly associated with AF” which would be a problem if they have used empirical variable selection, as it sounds like they have, but also this doesn’t tell the reader what was included in their model. The same point applies to “all other independent variables associated with AF” for Rashid Nadeem’s study. How did you decide between “age (<64, 65–74, and >75)” for Wei-Syun Hu’s study in Table 1 as this would provide two ORs but there was no overlap between the study IDs for >65 and >75 in Figure 2?

While I appreciate the removal of “strongest” for associations involving the independent variables, I’m still not completely convinced by the OR of 1.5 and above threshold (e.g. Line 48) for continuous variables (age here) that have been categorised. Perhaps as a better illustration, there is presumably some reference and non-reference age groups with an OR < 1.5 and others with an OR > 1.5 (including the two presented in Figure 2). So “age” (Line 49) by itself doesn’t have an OR of > 1.5, but “age over 65” would qualify (as you use on Lines 56–57 and 226–227, as examples). Related to this, it’s not clear to me what “education” in Figure 2 means. Is this completing high school versus not, per year of formal education, university/colleague education versus not, or something else? (I noted that “education” was mentioned in two of the studies in Table 1.) What does “Low HRQoL” mean in operational terms (I appreciate its semantic meaning, but what is the line between low and not-low HRQoL here)?

Also, if an OR > 1.5 is of interest, presumably ORs < 0.66… would also be of interest (as this depends only on the reference/non-reference group choice). I think it would be useful to add this around Lines 48, 165, 226, and 253.

You have very helpfully provided the reference categories for some variables (race, sex, urbanity, etc.) in Figure 2 (and for all relevant variables in Table 2). Can you add these for age (even if it is trivial for the age groups provided), education and Low HRQoL in Figure 2, along with some explanation of what they mean in the methods (HRQoL is mentioned by name on Line 254, but without explanation, and not listed in Table 1) and education is only listed in Table 1 without any further explanation.

I wasn’t able to see how the study IDs in Figure 2 linked to the actual studies. My apologies if I’m overlooking the obvious here.

Education is apparently from [16] only but there are two studies in Table 1 with education included (Kuang Ming Liao and Tomasz Rusinowicz). There are many studies in Table 1 with age as a covariate (18 I think), but only four in Figure 1, why is that? Similarly there are 16 with sex, but only 3 in Figure 1, which seems a very low rate to have included in the meta-analysis. Three studies mention smoking (one under “tobacco consumption”) and three mention BMI in the “Adjustment” column, what were the reasons for why these factors could not be looked at more closely? If these are due to the data extraction requirements, this seems important as a limitation to be added to the discussion.

---

## Round 0.3 · Minor Revisions

Still pending some minor issues to be taken into account in a new revised version of the text.

·

Basic reporting

Thank you for your thoughtful responses to my questions and the constructive revisions to your manuscript. I have a number of mostly minor queries and suggestions based on the revised manuscript.

Line 51: Remember that additivity (where +50% would be the inverse of -50%) is on the log-scale and not the exponentiated/ratio scale. If the reference group had odds of 0.10 and the comparison group had odds of 0.15, this would be an OR of 0.15/0.10 = 1.50 (and so meet this criterion if the inequality was non-strict). However, if you reverse the reference/comparison groups, leaving the magnitude of the effect unchanged, the OR would then be 0.10/0.15 = 0.667 (3dp), a 33.3% (3dp) decrease in odds rather than a 50% decrease. The inverse of a 50% increase (Line 48) or an increase of one-half would be a one-third decrease and not a 50% decrease (c.f. Line 51). See also Lines 169–170 and 266 for the same point in the main text.

Lines 83–84: I looked at Mapel, et al. 2005 for these figures and the closest I could find for the 14.3% stated here was “Among COPD patients hospitalized in 1998, the prevalence of coronary artery disease, congestive heart failure, and atrial fibrillation were very high (33.6%, 24.4%, and 14.3%, respectively)” which given their focus on veterans didn’t seem to align with “For patients with COPD, the worldwide prevalence…” (Lines 82–83), is for those hospitalised only, and didn’t specify “new-onset” (Line 83, for which shouldn’t “prevalence” on Line 83 be “incidence”, c.f. Lines 85–86?). I couldn’t find a mention of the 40% mentioned on Line 84, but perhaps this is in a table or figure that I’m overlooking or you derived it from values in the article. See also Line 301 for the same 40% and Line 300 for a 17% overall prevalence (which I also couldn’t find in Mapel, et al. 2005 which is included here as a reference alongside Patel et al.). Also the journal title on Line 490 should be “COPD” and not “Copd”.

Line 88: Perhaps related to the above, I couldn’t find the 5.7% mortality in Mapel, et al. (or the 2.2%, but I’m assuming this is from the other reference?) despite the reference on Lines 89–90. If I’m not overlooking anything here, it might be worth checking that each reported number in the introduction and discussion can be located in the reference(s) provided. My sincere apologies if I’m overlooking these values in the references.

Line 136: Rather than “inconsistency”, perhaps “disagreement” here?

Lines 136–137: While consulting the third reviewer is a good step, it doesn’t in itself guarantee that the disagreement would be resolved. Perhaps “…would be consulted to resolve the disagreement.” You state this more explicitly on Line 142 for data extraction, so this edit would make the text for study selection more similar to that.

Line 137: I think you mean “inter-rater” here (i.e., between raters, as stated on Line 138) rather than “intra-rater” (i.e., within a single rater). You use the correct “inter-rate” on Line 195.

Line 137: Perhaps add “analysis” here to read: “An inter-rater reliability ANALYSIS was conducted to”, or “Inter-rater reliability was CALCULATED to…”

Line 153: Perhaps “used” rather than “conducted” here.

Line 155: Is “all authors” here referring to “J Liu and J Lu” (Line 151) plus all seven other authors? That seems a very large number of authors to resolve inconsistencies.

Lines 158–159: I’m not entirely sure what this sentence is intended to communicate in terms of the options chosen in Stata.

Line 160: Spurious “in” in “in the in”.

Line 165: Perhaps “used” rather than “prepared” here.

Line 171: Perhaps “95% CIs were used…of the ORs.”.

Lines 171 and 172: Perhaps add “A”/”a” before “[w]ide CI”, “narrow CI”, and “95% CI spans” here.

Lines 172–173: As explained previously, this “rule” doesn’t apply when the standard error differs between CIs (based on observed values) and hypothesis tests (based on null values). On Line 257, you say that “All of the un-pooled ORs showed the 95% CI and P value with consistency.” which I interpreted as meaning that the sentence here (Lines 172–173) was satisfied in Figure 2. However, note the presence of some “very near misses” in the pooled analyses, in particular short acting b-agonists (where the limit is 1.00–1.71 but the p-value is not sufficiently close to 0.050, being 0.040, to be explained by rounding) and long-acting b-agonists (where the CI is 0.96–2.34 and so the p-value should be just above 0.05 rather than the reported p=0.646) and violations in the unpooled analyses for venous thromboembolism (95% CI 1.17–1.44, so not including 1.0, but p=0.092) and OSA (95% CI 0.85–5.69, so including 1.0, but p<0.001).

Lines 196–200: Since you list the references for these 20 studies immediately below (Lines 200–207) broken down by study type, this initial listing of the references seems unnecessary.

Line 216: “and ONE scoring six of nine”.

Line 222: In all but here and one other instance (Figure 2 abbreviations), you have a space between “95%” and “CI”.

Lines 229, 237, and 337: Here you use beta for beta-agonists, but note you use “b” in Figure 2.

Lines 251–252: Here you give only the sensitivity analysis I-squared statistic, but on Lines 253 and 254 you give both the overall and sensitivity analysis values.

Lines 253 and 254: Here you only give the sensitivity analysis ORs, but on Line 252 you give both the overall and the sensitivity analysis ORs.

Line 268: Should the “+” symbol be “and” or “alongside”? See also Figure 2.

Line 286: Missing comma between “75” and “acute” in “age over 75 acute care encounter”.

Lines 332 and 333: This is perhaps stylistic, but the “1” in FEV1 is usually a subscript.

Lines 361–362: I think readers will appreciate a short statement here about the effects of removing Desai et al. as a sensitivity analysis (summarising Lines 248–255). Note that “Appendix 3. Sensitivity analysis” doesn’t seem to show these sensitivity analyses, but rather appears to show the individual studies. I’m not confident that I’m interpreting these supplementary figures correctly so perhaps more explanation can also be added to the supplement as well.

Line 371: You say “intermediate variables” here and on Line 377 you say “mediators”. Either is fine, but perhaps the reader will find this easier to follow if you use the same term in both places.

Table 1: Note (b) about education could delete “The”.

Experimental design

No comment.

Validity of the findings

No comment.

Additional comments

No comment.

---

## Round 0.4 · Minor Revisions

Still pending some minor modifications suggested by one of the reviewers.

·

Basic reporting

See below.

Experimental design

See below.

Validity of the findings

See below.

Additional comments

Thank you for your very constructive responses to my queries and your thoughtful revisions.

I really have only one remaining query, albeit in two parts. First, you have added “Studies that reported ORs with 95% CIs and P values showed inconsistency were excluded.” (Lines 132–133) as part of your exclusion criteria, but this isn’t entirely clear in its intention. Do you mean that these were excluded on the basis that they were likely to have problems with their analyses and/or reporting (leading to the inconsistency) or something else?

Related to this, there are several reasons for why CIs and p-values can give (slightly) different interpretations of results in some cases when we move away from general linear models. As a trivial example, “csi 30 20 20 30, or” in Stata will show a RR (which for a rare outcome approximates the OR, although we are usually interested in the opposite direction for this approximation) where the 95% CI includes 1.0 (being 0.997, 2.256) but the p-value is below 0.05 (being 0.046). The OR has a 95% CI that is compatible with the reported p-value (being 1.016, 4.982 for Cornfield’s interval; or 1.011, 5.008 for Woolf’s interval), but the RR and its CI limits can of course be converted to ORs using the observed or an assumed prevalence where the incompatibility will again exist. Adding the exact option will report a two-sided Fisher’s exact test p-value (0.0713), which is compatible with the RR CI but not the OR CIs. In short, there are multiple ways of obtaining 95% CI limits for ORs (including Wald-based and from profiling the likelihood) and multiple ways of obtaining p-values (including those from Wald, LR, and score tests). Other issues can arise for more complex models, particularly, I believe, those involving weights and/or non-linear associations. Unless you wanted to go into detail and provide references for the potential sources of these inconsistencies, and I’m not aware of specific research on this issue for meta-analyses, I suggest deleting the text “In binary outcomes, however, when the standard error differs between CIs (based on observed values) and hypothesis tests (based on null values), it is possible to generate data with the 95% CI and P value showed inconsistency. This kind of situation basically does not happen in meta-analysis, as it can be avoided in original studies.” With the resolution of this issue in Table 2, this seems much less important to the reader now in any case. Of course, you could also expand on this section instead, but I suspect that this would become more of a distraction to readers.



I will also note a few small language issues.

Line 46: Here you say “during COPD”, where I think you mean “in COPD patients” (as you use on Line 44 and elsewhere), or perhaps “for those with COPD”. See also Lines 54, 55, 56, 61, 66, 97, 101, 229, 285, 286, 292, 298, 303, 313, 346, 402, and 408, and the caption for Figure 2 (and possibly elsewhere).

On Line 84 you give a category of “60–80%” % predicted FEV1, and on Line 85 you give FEV1 ≥ 80% % predicted as a separate category. Based only on this text, it’s not clear which category exactly 80% would belong to. The same point arises again on Lines 86 and 87 (although the “<60%” on Line 86 is entirely clear about the natural of the interval) and Lines 342 and 343.

Line 110: “referred” should be “Preferred” (being the “P” in “PRISMA”)

For Line 140, I suggest changing this to refer to the “the third reviewer (J Liu)” (as you did before) rather than saying that “it would be consulted” (it’s unclear what “it” is here).

In the revision to Lines 172–173, the added sentence: “As the inverse of a 50% increase in odds would be a 33.3% decrease in odds.” is a fragment (not a complete sentence). This could be corrected by removing “As”, by simply connecting it with the previous sentence without the current period (and either with or without a comma) and capitalisation, or the same but as a parenthetical remark.

Line 365’s “indicates” is a very strong word to use for results from observational studies. Perhaps “suggests” or similar would be more appropriate. It might be worth checking for any other overly strong causal language throughout the manuscript.

Line 369: “response” should be “respond”.

Line 370: “source” would read better as “sources” (there could be more than one).

Line 403: I suggest deleting “related to” so this reads “The dominant factors are cardiovascular (CAD, HF and CHF)…” (or you could reword this in another way.)

---

## Round 0.5 · accepted · Accept

All the reviewers' concerns have been correctly addressed.

·

Basic reporting

No further comments.

Experimental design

No further comments.

Validity of the findings

No further comments.

Additional comments

Thank you for your revisions and responses. I have no additional comments to make about your manuscript.